# Survival of Radioresistant Bacteria on Europa's Surface after Pulse Ejection of Subsurface Ocean Water

**Anatoly Pavlov [1], Vladimir Cheptsov [2,3,\*], Denis Tsurkov [1], Vladimir Lomasov [4], Dmitry Frolov [1,4] and Gennady Vasiliev [1]**

1   Ioffe Physical-Technical Institute, Russian Academy of Sciences, St. Petersburg 194021, Russia; anatoli.pavlov@mail.ioffe.ru (A.P.); d.tsurkov@me.com (D.T.); undead.rus@gmail.com (D.F.); gennadyivas@gmail.com (G.V.)
2   Department of Soil Biology, Lomonosov Moscow State University, Moscow 119991, Russia
3   Space Research Institute, Russian Academy of Sciences, Moscow 117997, Russia
4   Peter the Great St. Petersburg State Polytechnic University, St. Petersburg 194021, Russia; lomasoff@yandex.ru
\*   Correspondence: cheptcov.vladimir@gmail.com; Tel.: +7-929-917-07-80

**Abstract:** We briefly present preliminary results of our study of the radioresistant bacteria in a low temperature and pressure and high-radiation environment and hypothesize the ability of microorganisms to survive extraterrestrial high-radiation environments, such as the icy surface of Jupiter's moon, Europa. In this study, samples containing a strain of *Deinococcus radiodurans* VKM B-1422$^T$ embedded into a simulated version of Europa's ice were put under extreme environmental (−130 °C, 0.01 mbar) and radiation conditions using a specially designed experimental vacuum chamber. The samples were irradiated with 5, 10, 50, and 100 kGy doses and subsequently studied for residual viable cells. We estimate the limit of the accumulated dose that viable cells in those conditions could withstand at 50 kGy. Combining our numerical modelling of the accumulated dose in ice with observations of water eruption events on Europa, we hypothesize that in the case of such events, it is possible that putative extraterrestrial organisms might retain viability in a dormant state for up to 10,000 years, and could be sampled and studied by future probe missions.

**Keywords:** astrobiology; Europa; ionizing radiation; microorganisms; radioresistance; accelerated electrons

## 1. Introduction

Jupiter's satellite, Europa, is believed to have a subsurface ocean that might be a potential environment for the existence of extraterrestrial life [1]. Astrobiological studies concerning environmental factors and potential habitats [2], proposing sampling end experimental strategies [3,4], have been recently published. As the high radiation and low temperature in this case are believed to be limiting factors of biological activity, thus constraining possible habitats to the bottom of the ice sheet, seafloor, and ocean [2], it seems crucial to study the effect of these factors on living organisms, simulating the harsh environment of Europa. Observational data from the Hubble Space Telescope (HST) provide possible evidence for water plumes being ejected from the ocean to space and permanently renewing the icy surface of Europa [5,6]. The ocean is hypothesized to be habitable. In this case, hypothetical microorganisms could be ejected from the ocean and frozen in the surface ice layer during the water eruption event. These potential life forms might remain in a dormant state for an undetermined extended period of time at a very low temperature in the surface ice layer.

The viability of microorganisms after millions of years of cryoconservation should not be challenging *per se*, because living cells are found in ancient permafrost rocks that have not melted for several Myr (e.g., [7–9]), but Europa's orbit being placed in the radiation belt of Jupiter means the moon's surface is subject to high doses of radiation [10]. This radiation is sufficient to sterilize the surface ice layer, however, our modelling results show that this effect sharply decreases with depth, mainly as a result of highly effective energy loss of MeV energy range electrons and ions in the first centimeters of ice. The determination of the survival time and depth limits is the main aim of this study. We performed irradiation (with accelerated electrons) of *Deinococcus radiodurans* bacteria embedded into a model of Europa's ice under simulated temperature and pressure conditions of Europa.

## 2. Materials and Methods

### 2.1. Vacuum Chamber

A special vacuum chamber was developed for modelling the extreme irradiation of icy samples with bacteria under low temperature and low atmospheric pressure ($-130\ °C$, 0.01 mbar).

The chamber is a cylindrical stainless steel tank covered thin Al film (100 μm thickness). High energy electrons pass through the film to the vacuum chamber and bombard samples placed on the bottom surface of the chamber. A number of cylinders were installed inside the chamber to support the film against external atmospheric pressure. The chamber was directly connected to a liquid nitrogen tank, which was placed below the chamber. The bottom part of the sample chamber was cooled during irradiation run by several "cold copper fingers" loaded into liquid nitrogen. The chamber was permanently pumped down to 0.01 mbar during the experimental run. The temperature on the bottom surface of the chamber was monitored with a thermocouple. Samples were put on the bottom of the chamber and had a thickness of about 1 mm, so that the irradiation of the sample was close to uniform. The scheme of the chamber is presented in Figure 1.

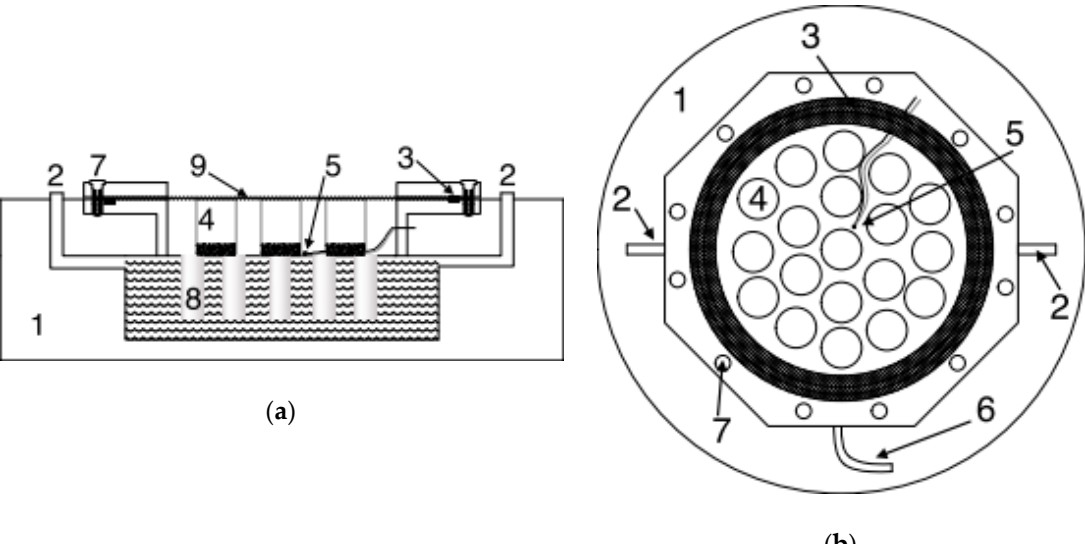

(**a**)

(**b**)

**Figure 1.** Scheme of the experimental chamber: (**a**) side view; (**b**) top view. 1—Thermoisolated cylinder, 2—tubes for liquid nitrogen filling, 3—vacuum rubber seal, 4—metallic cylinders supporting the Al foil, 5—thermocouple, 6—tube for vacuum pumping, 7—metallic ring with bolts for vacuum seal, 8—copper "cold fingers", 9—Al foil with thickness 100 μm.

### 2.2. Accelerator

Irradiation was produced by the electron accelerator with electron energy in the range of 0.4–1.0 MeV and beam intensity in the range of 0–3 mA. The electron beam exit window has a length of 450 mm and a width of 20 mm. The vacuum chamber on the mobile platform was placed

under the exit window. The platform then moved forward and backward to obtain uniform irradiation of the whole area of samples (see Figure 2).

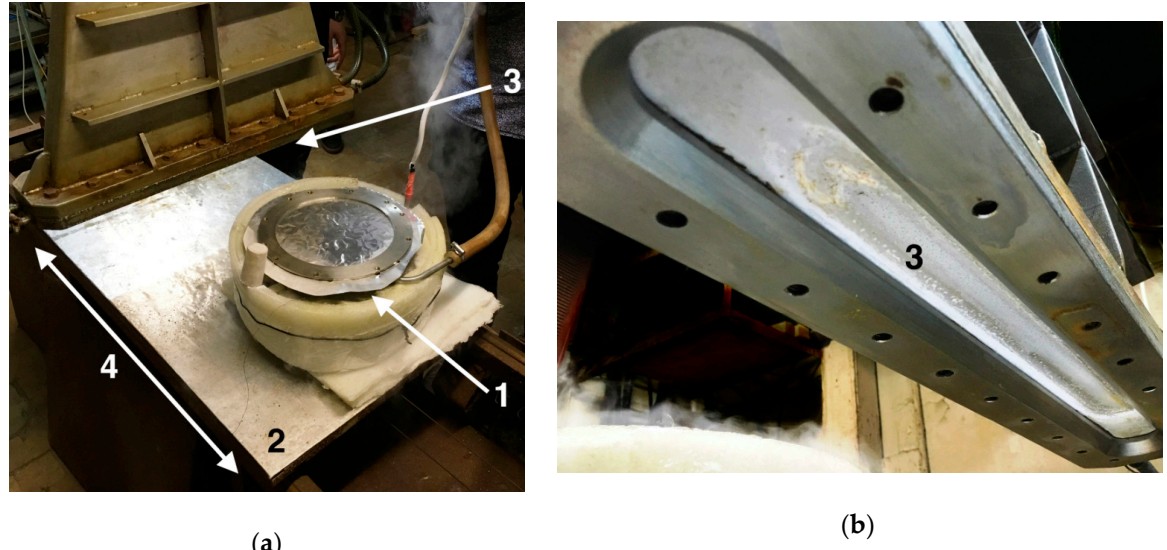

(**a**)          (**b**)

**Figure 2.** Climatic chamber and exit window of accelerator: (**a**) chamber on mobile platform under the exit window of accelerator; (**b**) exit window of accelerator. 1—chamber, 2—mobile platform, 3—exit window of accelerator, 4—direction of movement of the platform.

### 2.3. Samples Preparation and Analysis

Strain *Deinococcus radiodurans* VKM B-1422[T] was the object of study. A solution of salts modeling the composition of Europa's ice was used for suspension of bacterial cells.

There are no direct measurements of Europa's ice composition, but several studies assume $Mg^{2+}$, $Na^+$, and $SO_4^{2-}$ to be the dominant ions within it [11–13]. We follow the composition described by the authors of [13] to prepare the model ice samples (artificial Europa ice with bacteria). A solution of magnesium sulfate, sodium sulfate, and sulfuric acid at a ratio of 50:40:10 of molar percent with 35 g/L total concentration was used.

To avoid contamination, the samples of ice with bacteria embedded in it were hermetically packed in polyethylene film. Each packed sample had a diameter of about 20 mm and a thickness of about 1 mm. The samples were put on the bottom of the chamber, which was pre-cooled to −100 °C. The film was pierced just prior to the chamber sealing and pumping so that the pressure inside the chamber and the sample containment would be equalized during the experiment. The irradiation run began after the stabilization of temperature and pressure in the vacuum chamber at around −130 °C and 0.01 mbar. Four runs of irradiation were executed with doses of 5, 10, 50, and 100 kGy. Irradiation rates were 0.28 kGy/s for 5 and 10 kGy runs and 2.8 kGy/s for 50 and 100 kGy runs. The time from placing the samples in the chamber to unloading from the chamber (including the time of air pumping out, the onset of temperature equilibrium, the exit of personnel from the room, the starting and stopping of the accelerator, and the irradiation) was about 10 min. After the irradiation, the samples were immediately taken out of the chamber and put into sterile 15 ml polypropylene tubes installed in a cooled thermos. All transport and store operations were carried at −18 °C temperature.

The determination of the number of culturable bacteria in the samples was performed by plating on glucose–peptone–yeast agar (GPY) (peptone—2 g/L, glucose—1 g/L, yeast extract—1 g/L, casein hydrolysate—1 g/L, CaCO3—1 g/L, agar–agar—20 g/L) [14]. Suspensions of samples in different dilutions were plated in triplicate under sterile conditions with simultaneous control of the nutrient medium sterility, sterility of the water used for preparation of dilutions, and control of the presence of foreign air microflora. The plates were incubated at a temperature of +28 °C for two weeks.

### 2.4. Calculating Accumulated Dose after Water Release on the Surface of Europa

A GEANT4 toolkit [15] was used first to calculate the energy absorbed in a unit water ice layer per incident particle with a certain total energy. GEANT4 was initially developed by CERN co-workers [16] for implementation in high-energy particle physics, and utilizes intranuclear and internuclear cascade models for particle interactions. It provides a wide set of tools for the simulation of particle interactions with matter, and the main advantage of using this toolkit is that it allows the consideration of not only the incident particle effects on the target, but also of contributions from secondary particles, with gamma-rays and X-rays included. The depth of the unit layer was considered to be 0.1 mm and its incident surface as 1 cm$^2$ with the total depth of the ice column being from 0.05 to 5000 mm. These absorbed energy/depth $E_{abs}$ $(E, d)$ profiles were calculated for electrons, protons, oxygen, and sulfur ions as incident particles. To imply these calculations to Europa's radiation environment, we used differential particle spectra $J$ $(E)$ [counts per cm$^2$ s sr MeV] from Table 1 in the work of [10]. To obtain total accumulated dose at each depth $d$, one has to numerically integrate $J$ $(E) \times E_{abs}$ $(E, d)$ $dE$ for every sort of incident particle, which results in a dose rate profile. Thus, we obtain a one-year dose profile.

A simple model of Europa resurfacing was introduced for the following calculations: A layer of "fresh ice" with varying thickness is added atop the ice profile and the calculation is then continued for further 1 million years. Ip et al. [17] proposed the static resurfacing rate of 12 m per 100 Myr, which was also taken into consideration. These computations were done with MATLAB software.

## 3. Results and Discussion

### 3.1. Bacteria Survival after Irradiation

After irradiation with doses of 5, 10, and 50 kGy, the number of cultivated cells decreased on 2, 3, and 6 orders of magnitude, respectively (Figure 3). Cultivated cells were not found in samples that were irradiated with the 100 kGy doses. Under the conditions of our experiment, *Deinococcus radiodurans* VKM B-1422$^T$ demonstrated radioresistance that significantly exceeded its radioresistance at normal conditions [18,19]. This is the result of the decrease of radiation damages under irradiation at a low temperature and low pressure. This effect was discussed by us previously in detail [14,20,21]. Our result is in general accordance with studies of radioresistance of *Deinococcus radiodurans* carried out previously with high doses of γ-ray irradiation at −79 °C [22,23]. However, there is a difference in the decrease of bacterial survival with escalation of the radiation dose. In γ-ray experiments, bacteria populations show signs of decrease only after 30 kGy dose accumulation, whereas in our high energy electrons experiment, such a process begins at 5 kGy. There might be two possible causes for this. It might be that the MeV-electrons produce deadly damages more effectively than γ-rays [20], or it could be the difference in composition of the ice samples used, because salts and minerals may be sources of free radicals at radiolysis (e.g., [24]). Authors of the abovementioned papers [22,23] used an ordinary nutrient solution, while we used a special solution simulating the composition of Europa's ice.

We should note that we studied pure bacteria culture, while in nature, microorganisms exist in communities, and inter- and intrapopulation interactions as well as interactions with the environment can significantly increase microbial radioresistance [20,25]. Thus, studies of natural microbial communities' radioresistance would be eligible to specify putative life survivability on Europa under the accumulation of radiation dose. Microbial communities of Earth glaciers could be proposed, like terrestrial analogs of Europa ice, for such studies [4,26,27].

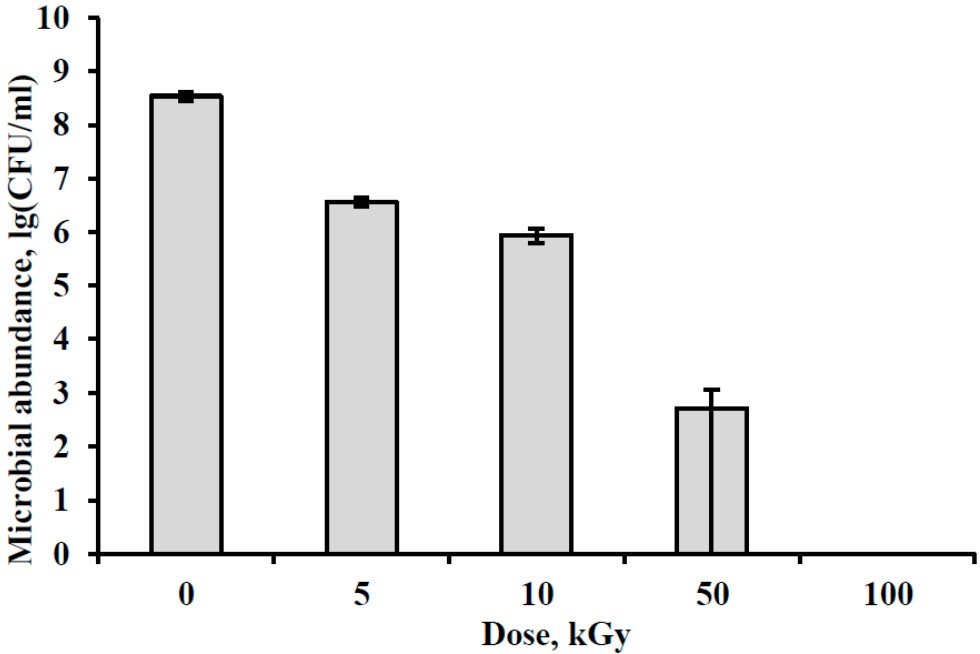

**Figure 3.** Impact of high energy electrons irradiation on the number of *Deinococcus radiodurans* VKM B-1422[T] viable cells. Error bars are in accordance with the confidence interval for $p < 0.05$.

It should be noted that the irradiation intensity used in the experiment was several orders of magnitude higher than on Europa (see Section 3.2). The effect of such differences on the microorganisms' viability was discussed by us in detail earlier [20,21]. Briefly, the effectiveness of a sparsely ionizing radiation (including accelerated electrons) dose is reduced with irradiation intensity decreasing as a result of recovery phenomena [28]. Under the low-temperature conditions of Europa, this phenomena should be insignificant because of the low rate or full stop of metabolism [29].

### 3.2. Dose Dependence on Depth and Exposure Time

The total of all dose accumulation rates of considered incident particles is presented in Figure 4. In our calculations, the resulting dose rates for different particles (Figure A1) were close to that of Paranicas et al. [10] at depths less than 10 cm, but for depths more than 10 cm, our modelling demonstrated higher dose accumulation values. Those differences increase with depth up to 10 times at 1 m. This is probably the result of using modern models of secondary particle cascade in our GEANT4 calculations. High-energy protons and heavy ions profiles dramatically decrease with depth, thus their impact on the microorganisms' survival in the ice layer is significant only for depths less than 1 cm, even taking into account their higher biological efficiency compared with electrons. Despite the high total dose accumulation rate, the lethal dose does not accumulate in the ice layer at depths of 0.1–1 m until 1000–10,000 years after eruption. For lengths of time greater than 10,000 years, the dose is lethal in the region of interest. For exposure times from 1000 years onwards, we see a nearly constant (in comparison with the one-year profile) part emerging in the dose profile. This is the effect of considering a static resurfacing rate from the work of [17] in our model. This effect might not be real case, but it has no significant impact on the main goals and conclusions of this work. At 0.01–0.1 m depths, the lethal dose is accumulated in 100 years, and for depths lower than 0.01 m, no viable cells would be present after only one year of radiation exposure.

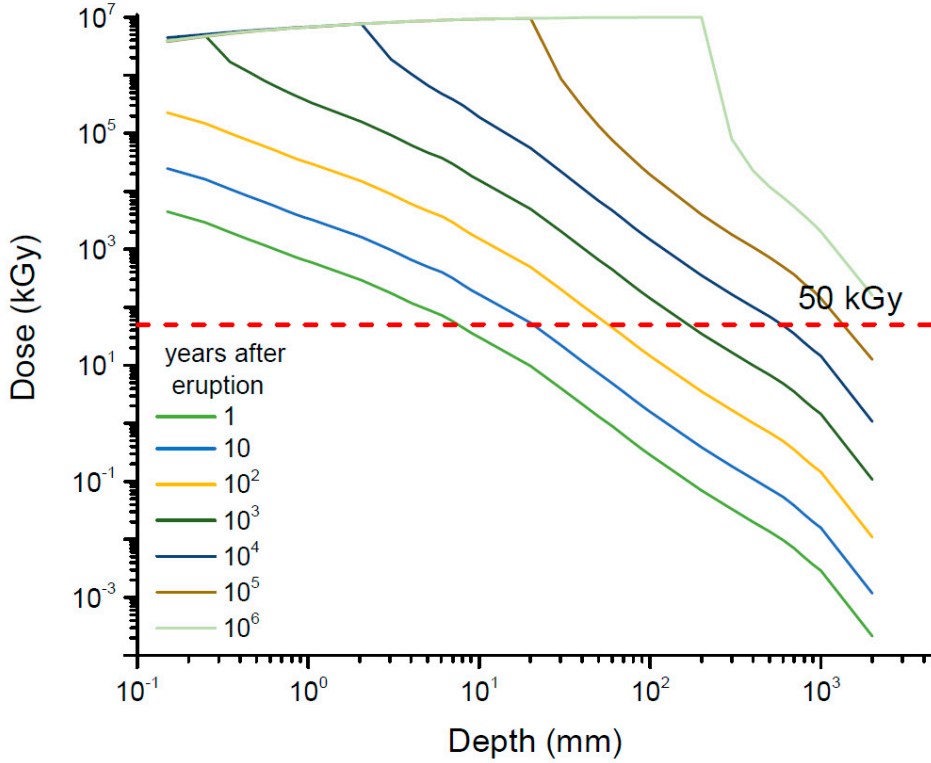

**Figure 4.** Total dose accumulation rate in dependence on exposure time and "fresh ice" depth. Red dashed line demonstrates the upper limit of the *Deinococcus radiodurans* VKM B-1422[T] survival in our experiment.

## 4. Conclusions

Our study demonstrates the ability of terrestrial microorganisms to survive high intensity radiation in conditions modelling shallow subsurface environments of Europa for a long time. It could be used in future missions to Europa aimed at finding hypothetical extraterrestrial life that has been ejected on the surface by sporadic water release. Considering radiation dose accumulation as the limiting factor for microbial life survivability, we hypothesize that there is a chance to discover viable cells at an ice depth of 10–100 cm if the massive water release from the ocean of Europa occurred on a landing site 1000–10,000 years before. Evidence for such water eruptions (plumes) on Europa has been previously found and reported by the HST team and other researchers (the works of [5,6] and references therein) and from these observations, the possible locations for probing and sampling might be determined. The events observed by these authors occurred at the same location with a separation of two years, thus potentially marking the source of "fresh ice" subsurface sample candidates on Europa.

**Author Contributions:** A.P., V.C., and D.T. designed the study; V.C., D.T., and V.L. performed the experiment; D.F. and G.V. performed model calculations; A.P., V.C., and D.F. prepared the draft manuscript; all authors edited and commented on the manuscript.

**Funding:** This research and the APC were funded by Russian Science Foundation, grant number 17-12-01184.

**Acknowledgments:** Authors thank Solovyov V.Yu. for the help in irradiation. The authors thank anonymous reviewers for the help in improving the article.

**Conflicts of Interest:** The authors declare no conflict of interest. The funders had no role in the design of the study; in the collection, analyses, or interpretation of data; in the writing of the manuscript; or in the decision to publish the results.

## Appendix A

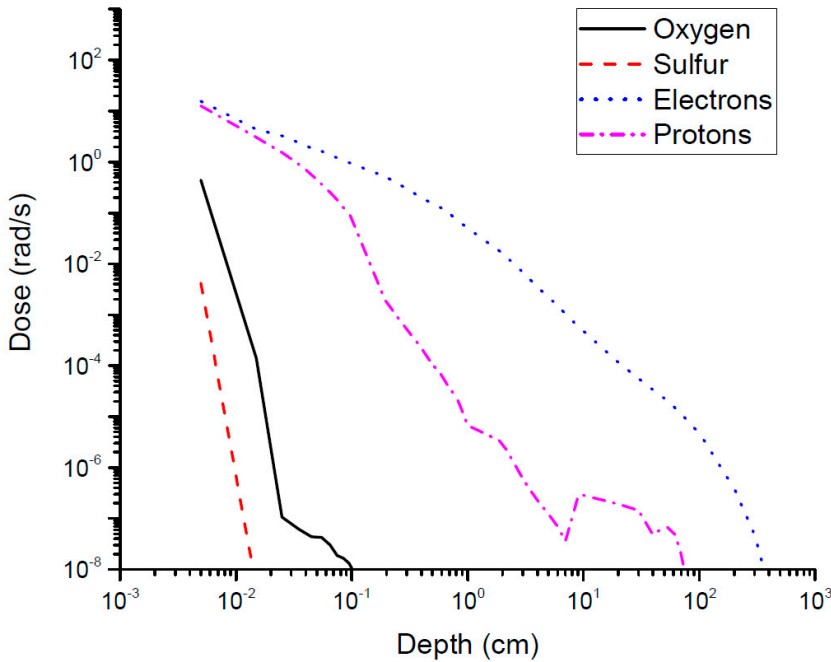

**Figure A1.** Calculated dose rates for incident particles.

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
