# Peer review of "Survival of Radioresistant Bacteria on Europa’s Surface after Pulse Ejection of Subsurface Ocean Water"

_geosciences, doi:10.3390/geosciences9010009_

Round 1
Reviewer 1 Report
See attached document.

Author Response
Dear Reviewer,
Thank you for your thorough review and for helping in improving of the manuscript! Please find our detailed response to reviewers below. For your convenience the questions are marked as “Q:” and our answers marked as “A:”, and questions and answers are written in black and red color, respectively.
Reviewer 1
The presented manuscript is interesting, deal with the recent scientific topic and probably will pay attention of a broad audience of scientists, especially astrobiologists. However, I see weak points which should be corrected and may improve a manuscript. Even though it is a short note, some parts of the manuscript should have a better background and be described more carefully as well.
Comments:
Q: Title: I suggest change the title. Conditions used in experiment do not reflect exactly conditions of Europa. Moreover, a new title may bring attention of more readers. Please focus in the title on icy moons. Microorganisms refers to a wide spectrum of microbes, even protozoans. Please, just say "bacteria".
A: We emphasize that the modelling conditions were chosen mainly similar to that on Europa.
· We study extremal radiation conditions, and radiation fluxes on Europa are the highest among all other icy moons
· We need suitable conditions for the suggested mechanism of ejecting water and fixing it back on the surface. E.g. on Enceladus water vapors diffuse after ejection due to the moon’s lower mass.
We though changed “microorganisms” to “bacteria” in the title, as you fairly suggested.
Abstract:
Q: Line 17: add in bracket value of temperature and pressure.
A: Corrected as suggested.
Q: Line 18: add mean value of radiation in brackets. It is not the best idea to hide basic variables.
A: The sentence is corrected: “The samples were irradiated with 5, 10, 50, and 100 kGy doses and subsequently studied for residual viable cells”.
Q: Line 23: Unfortunately, in a fast-moving scientific world people used to read only abstracts. Please, specify in the text: (a) that you used special designed chamber, (b) are you able to say how long time bacteria might survive? (c) are you talking about active/viable bacteria? At the same beginning of abstract, it is worthy to say that Earth is not only celestial body covered by water ice.
A: а) It is stated on line 18
b) We changed the sentence to «it is possible that hypothetical extraterrestrial organisms might retain viability in a dormant state up to 10000 years»
c) We mean viable bacteria, so we are changed the formulation to “viability”.
Concerning the commentary about extraterrestrial ice, we believe it is not a crucial statement, as it is wide known that water ice is found in comets, on Mars, recently, and Enceladus also. If you though believe this statement to be essential for the abstract, we are ready include it to the MS.
Introduction:
Q: Line 33: in this case, you do not have to say "microorganisms", just say organisms since we do not know what live there (if something...)
A: Well that might be true, but we studied the survivability of bacteria and are very careful with our suggestions as they already might be seen as highly speculative. Thus, we believe we should keep the “micro-” part.
Q: Line 34: "can then remain" - at this moment we do not know whether life is present or absent beyond Earth, thus you can not say "can". It is much friendly to sceptics like me to hear "potential life forms might remain..."
A: Corrected as suggested: “These potential life forms might remain in a dormant state for an undetermined extended period of time at very low temperature in surface ice layer.”
Q: Line 39: could you give an example of such decreasing on Earth's ice? I think about electrons.
A: Unfortunately, this is impossible on Earth due to the thickness of its atmosphere.
General comment to Introduction:
Even though it is a short note, this Introduction is too short...
Please, add a few sentences that astrobiological studies devoted to Europa are recently published. It is worth to say that inspiration for searching of life on icy moons are glacial ecosystems on Earth, which despite harsh conditions are inhabited by myriad of bacteria, protozoans and even animals. I suggest to give background that terrestrial glaciers migh be analogues and they are not simple but complicated ecosystems e.g.:
1. Hodson A., Anesio A.M., Tranter M., et al. (2008) Glacial ecosystems. Ecological Monographs 78: 41-67.
2. Takeuchi N., Kohshima S.S., Seko K. (2001) Structure, formation, and darkening process of albedo-reducing material (cryoconite) on a Himalayan glacier: a granular algal mat growing on the glacier. Arctic, Antarctic, and Alpine Research 33: 115-122.
3. Zawierucha, K., Ostrowska, M. & Kolicka, M. (2017) Applicability of cryoconite consortia of microorganisms and glacier-dwelling animals in astrobiological studies. Contemporary Trends in Geoscience, 6(1): 1-10.
A: Regarding recent astrobiological studies devoted to Europa, some sentences are added: “Astrobiological studies concerning environmental factors and potential habitats [2]; proposing sampling end experimental strategies [3,4] are recently published. As the high radiation and low temperature in this case are believed to be limiting factors of biological activity, thus constraining possible habitats to the bottom of the ice sheet, seafloor and the ocean [2], it seems crucial to study the effect of these factors on living organisms, simulating the harsh environment of Europa.”
Regarding glaciers, we have mentioned it now, but we assume that detailed discussion of it will be inappropriate due to several reasons:
a) There are several types of terrestrial analog ecosystems for Europa - not only glaciers, but also subsurface lakes, permafrost, cryopegs etc. So it will be strange to thoroughly discuss only glaciers, while detailed discussion of all analogs will be excessive and will tangle the idea of present research, which is rather clear in current view.
b) We consider water ejection from subsurface ocean, so subsurface lakes would be even more appropriate analogs for our study than glaciers.
c) We heavily support the idea of using of natural ecosystems in model astrobiological experiments (see, e.g., Cheptsov et al., 2017; Cheptsov et al., 2018a; 2018b cited in the MS). But in current research we studied pure culture, because it is preliminary research, and it is first experiment which simultaneously models pressure, temperature, radiation and salts in Europa ice. So, it would be contradictory to discuss structure of glaciers’ microbial communities’ structure and usage of this like analogs, while we used pure culture.
Nevertheless, we are added some related sentences: “We would note that we studied pure bacteria culture, while in nature microorganisms exist in communities, and inter- and intrapopulation interactions as well as interactions with environment can significantly increase microbial radioresistance [20,25]. Thus, studies of natural microbial communities’ radioresistance would be eligible to specify putative Europa life survivability under accumulation of radiation dose. Microbial communities of Earth glaciers could be proposed like terrestrial analogs of Europa ice for such studies [4,26,27].”
Methods:
Q: Line 46: specify preassure and temperature
A: Corrected as suggested.
Q: Figure 1: the lack of no. 5 on figure (a) and lack of 8 on (b) is misleading. Please add.
A: No. 5 is added on figure 1a. Unfortunately, we are unable to add No. 8 on figure 1b because this detail is not visible in top view.
Q: Figure 2: could you add arrowheads to make clear for readers what you would like to describe? I mean "exit window". Is it the same chamber as described above (Figure 1)? if yes please, could you make a picture of this chamber too? Sheme in Figure 1 is really good, but the additional picture will make paper more professional.
A: Arrowheads are added on figure 2, and additional figure 2b is added. We hope it will clarify our description. Moreover, probably misunderstanding occurred due to our bad wording, so we have rephrased the sentence in lines 82-83 like “Electron beam exit window has 450 mm length and width of 20 mm.”
Q: Line 68: Authors write about microbiological analysis. I reccomend carefully describe that everything was done under sterile conditions using gloves etc....if not say about it.
A: Initially sterility controls were described in lines 119-122: “Suspensions of samples in different dilutions were plated in triplicate with simultaneous control of the nutrient medium sterility, sterility of the water used for dilutions preparation, and control of the presence of foreign air microflora.” In such works, compliance with sterility is implied, and the details you specified are usually not specified. Nevertheless, we have modified the sentence and added the phrase “under sterile conditions”.
Q: Lines 70-71: please, rephrase this sentence. You can not say that "bacteria biomass was suspended...", rather say solution of....was used for suspension of bacterial cells
A: Corrected as suggested: “Solution of salts modeling composition of Europa ice was used for suspension of bacterial cells.” – lines 97-98.
Q: Lines 72-74: It is not clear. I'm not sure what you have done. If I well understand you prepare artificial Europa ice with bacteria. I recommend describe it more carefully and make clear for readers.
A: To clarify, we have rewritten this part of the text: “Strain Deinococcus radiodurans VKM B-1422T was the object of study. Solution of salts modeling composition of Europa ice was used for suspension of bacterial cells.
There are no direct measurements of the Europa’s ice composition but several studies assume Mg2+, Na+ and SO42- as dominant ions in it [11-13]. We follow the composition described in [13] to prepare the model ice samples (artificial Europa ice with bacteria). Solution of magnesium sulfate, sodium sulfate and sulfuric acid at ratios 50:40:10 of molar percent with 35 g/l total concentrations was used.
To avoid contamination, the samples of ice with bacteria embedded in it were hermetically packed in polyethylene film.”
Q: Line 82: how long time each treatment was conducted?
A: The next sentence is added: “The time from placing the samples in the chamber to unloading from the chamber (including the time of air pumping out, the onset of temperature equilibrium, the exit of personnel from the room, the starting and stopping of the accelerator, and the irradiation) was about 10 min.”
Q: Line 108: mention software for calculation
A: The next sentence is added: “These computations were made with MATLAB software.”
Results and discussion:
Q: Line 125: I like these results and they are very interesting but below authors mentioned about depth of ten centimetres of ice, what about other layers and viable bacteria there?
A: Please refer answer to your next question, please.
Q: Line 136: what about layer 0-0.1 m.?
A: Paragraph 3.2 was rewritten heavily, but we also added a commentary at lines 224-226 to address your comments: “At 0.01 – 0.1 m depths lethal dose is accumulated in 100 years, and for depths lower than 0.01 m no viable cells would be present after only 1 year exposure.”
Conclusion:
Q: Line 144: If you will give references how long time bacteria can survive frozen, this sentence will be more "empirical". At this moment it is only speculation. Please, clarify or delete. This time is too long I think. But maybe I’m wrong. Please, refer some papers.
A: Appropriate comment is added in Introduction (lines 39-41): “Viability of microorganisms after millions of years of cryoconservation should not be challenging per se, since living cells are found in ancient permafrost rocks not melted for several Myr (e.g. [7-9]), but Europa’s orbit being placed in the radiation belt of Jupiter makes moon’s surface subject to high doses of radiation [10].”
General comment:
Q: I feel that authors should say that on Earth, are more organisms (even multicellular) which are able to survive in extreme ecpsystems and may serve as a good astrobiological models. Glaciers constitute analogues and it is worth to underline it. It is a good for the beginning of discussion. Maybe it will be inspiring for others scientists. Below list of papers of people interested in astrobiology, and you can see that on Earth are hypothetical systems and organisms able to survive hostile conditions. Worthly to mention.
Authors discuss biogeochemistry of Antarctic cryoconite holes as astrobiological analogues:
1. Tranter M., Fountain A., Fritsen C., et al. (2004) Extreme hydrological conditions in natural microcosms entombed within Antarctic ice. Hydrological Processes 18: 379-387.
In this paper authors rcovered animals after beeing frozen for 11 years. They discuss that black pigmentation and other physiological adaptations allow them to survive in high mountain ice:
2. Zawierucha, Z., Stec, D., Lachowska-Cierlik, D., Takeuchi, N., Li, Z. & Michalczyk, Ł. (2018) High mitochondrial diversity in a new water bear species (Tardigrada: Eutardigrada) from mountain glaciers in central Asia, with the erection of a new genus Cryoconicus. Annales Zoologici, 68(1): 179-201.
Excellent paper which should also be mentioned. Cryogenian period and snowball earth has implications to your observations:
3. Vincent W.F., Howard-Williams C. (2000) Life on snowball Earth. Science 287: 2421.
A: Please refer to our previous answer, related to your General comment to Introduction.
Q: I hope these comments will help you improve manuscript. I feel it is worth to publish and is really interesting but needs more efforts. Thank you for inspiring research.
A: Thank you for your thorough review and for helping in improving of the manuscript. We hope that we have well addressed all your questions and comments, and the manuscript can be published in its present form. Otherwise, we are ready to continue improving of the MS.
Reviewer 2 Report
I thought that the ideas in the manuscript were clearly laid out and interesting, although there are grammar issues throughout that should be addressed. I have only a few points that could improve the manuscript:
-From the methods, I thought that the authors would model radiation doses of incident heavy ions in addition to electrons, but it looks like Fig. 4 presents model results for only incident radiation of electrons. The authors should clarify if this is the case or not. They should also include results for different radiation types, particularly for heavy ions, given that heavy ions are a major component of radiation around Jupiter.
-Surfaces on Europa could be quite a bit older than 10,000 years. I recommend that the authors have their model irradiation runs extend to ~100 Ma, which is the approximate average age of Europa's surface.
-This is just a suggestion, but have the author's thought about the probable formation of aqueous glasses during eruptions on Europa? This has relevance because it would aid in the preservation of microbes. Given the salt compositions used, and my own experience in forming aqueous glasses, the described freezing procedure probably produced glasses. See the following for recent research on this:
Toner, J.D., Catling, D.C. and Light, B., 2014. The formation of supercooled brines, viscous liquids, and low-temperature perchlorate glasses in aqueous solutions relevant to Mars. Icarus, 233, pp.36-47.
Author Response
Dear Reviewer,
Thank you for your thorough review and for helping in improving of the manuscript! Please find our detailed response to your comments below. For your convenience the questions are marked as “Q:” and our answers marked as “A:”, and questions and answers are written in black and red color, respectively.
Reviewer 2
Q: (x) Moderate English changes required
A: English editing is performed. In particular, section 3.2 is rewritten. If English is still unsatisfactory, we are ready to perform extra editing.
Q: I thought that the ideas in the manuscript were clearly laid out and interesting, although there are grammar issues throughout that should be addressed. I have only a few points that could improve the manuscript:
-From the methods, I thought that the authors would model radiation doses of incident heavy ions in addition to electrons, but it looks like Fig. 4 presents model results for only incident radiation of electrons. The authors should clarify if this is the case or not. They should also include results for different radiation types, particularly for heavy ions, given that heavy ions are a major component of radiation around Jupiter.
A: On Figure 4 we present the total of all dose accumulation rates for electrons, protons and heavy ions. Probably we made it unclear. We have changed the figure caption and corrected the manuscript:
“Figure 4. Total dose accumulation rate in dependence on exposure time and “fresh ice” depth. Red dashed line demonstrates upper limit of the D. radiodurans VKM B-1422T survival in our experiment.”
“The total of all dose accumulation rates of considered incident particles is presented on Fig. 4. In our calculations resulting dose rates for different particles (Figure A1) were close to that of Paranicas et al. [10] at depths less than 10 cm, but for depths more than 10 cm our modelling demonstrated higher dose accumulation values.” – section 3.2.
Also the figure including the results for different radiation types is added as Appendix.
Q:-Surfaces on Europa could be quite a bit older than 10,000 years. I recommend that the authors have their model irradiation runs extend to ~100 Ma, which is the approximate average age of Europa's surface.
A: On Figure 4 we added the calculated curves for 100 kyr and 1 Myr and made appropriate corrections in the text of the MS. We shortly discuss (in section 3.2) the effect of static resurfacing rate on accumulated dose profile. We did not consider greater depths and times as the aim of this paper is to advice the future mission on sampling site, and sampling from depths greater than 1 m, we believe, is technically very challenging.
Q:-This is just a suggestion, but have the author's thought about the probable formation of aqueous glasses during eruptions on Europa? This has relevance because it would aid in the preservation of microbes. Given the salt compositions used, and my own experience in forming aqueous glasses, the described freezing procedure probably produced glasses. See the following for recent research on this:
Toner, J.D., Catling, D.C. and Light, B., 2014. The formation of supercooled brines, viscous liquids, and low-temperature perchlorate glasses in aqueous solutions relevant to Mars. Icarus, 233, pp.36-47.
A: Unfortunately, aqueous glasses, described in the abovementioned paper, could not be modeled in our experimental setup. Our samples were thin so the irradiation would be uniform with depth. Also the pressure and temperature range in the experimental chamber seem to be out of the range at which these glasses form, as our samples were placed on a metal surface which was in direct contact with liquid nitrogen at 0.01 mbar pressure. Thank you for the suggestion, we will take it into account in our future work.
Reviewer 3 Report
Dear authors,
I have read with a great attention your manuscript entitled “Survival of radioresistant microorganisms on Europa’s surface after pulse ejection of subsurface ocean waters” submitted for publication in Geosciences.
As a general comment, your paper is interesting and well-written; however, I have some difficulties with the relevance of your results with respect to the viability of microorganisms. Indeed, you did not take into account the notion of time. The fact to receive a dose equivalent of several thousand years in only a few tens of seconds (according to your data l. 81-82) is dramatically different for living systems, even in a dormant state. Of course, I understand that it is not possible to carry out your experiments over several years; nevertheless it could be interesting, for a given dose, to make irradiations at different rates. At least, I think it is important to discuss this aspect in your manuscript, in particular in your conclusion (one or two sentences would be enough). Similarly, in the sentence l.143-144, I think it is difficult to determine if microorganisms would still be viable after 10000 years, even without irradiation. As a small comment, all your paper is well written except the paragraph 3.2.
Otherwise, I think your manuscript is interesting and suited for publication in Geosciences.
Best regards
Author Response
Dear Reviewer,
Thank you for your thorough review and for helping in improving of the manuscript! Please find our detailed response to your comments below.
Reviewer 3
Comments and Suggestions for Authors
Dear authors,
I have read with a great attention your manuscript entitled “Survival of radioresistant microorganisms on Europa’s surface after pulse ejection of subsurface ocean waters” submitted for publication in Geosciences.
As a general comment, your paper is interesting and well-written; however, I have some difficulties with the relevance of your results with respect to the viability of microorganisms. Indeed, you did not take into account the notion of time. The fact to receive a dose equivalent of several thousand years in only a few tens of seconds (according to your data l. 81-82) is dramatically different for living systems, even in a dormant state. Of course, I understand that it is not possible to carry out your experiments over several years; nevertheless it could be interesting, for a given dose, to make irradiations at different rates. At least, I think it is important to discuss this aspect in your manuscript, in particular in your conclusion (one or two sentences would be enough). Similarly, in the sentence l.143-144, I think it is difficult to determine if microorganisms would still be viable after 10000 years, even without irradiation. As a small comment, all your paper is well written except the paragraph 3.2.
Otherwise, I think your manuscript is interesting and suited for publication in Geosciences.
Best regards
Answer:
Dear Reviewer,
Thank you for your kind suggestions. Regarding high irradiation intensity in experiment comparing with dose rate on Europa, the issue was discussed by us in detail in our previous papers (Cheptsov et al., 2018a; 2018b). But we are agree, that it should be mentioned in present MS due to importance of the issue. It is known, that in general the effectiveness of a sparsely ionizing radiation (including accelerated electrons) dose is reduced with irradiation intensity decreasing (Baumstark-Khan, Facius, 2002). Usually it is explained by recovery phenomena. In the earlier study on bacteria survival under gamma-irradiation at −79 °C several radiation rates were used, but there was not clear differences observed (Dartnell et al., 2010). It can be explained by low rate or full stop of metabolism under low temperature conditions (Rummel et al., 2014). Nevertheless, recently we have performed some experiments to study the issue (namely we made irradiations of samples at different rates) and we are in the process of the samples analysis. Appropriate comments are added to the text of the MS:
“It would be noted, that the irradiation intensity used in the experiment was by the several orders of magnitude higher than on the Europa (see section 3.2). Effect of such differences on the microorganisms’ viability was discussed by us in detail earlier [20,21]. Briefly, the effectiveness of a sparsely ionizing radiation (including accelerated electrons) dose is reduced with irradiation intensity decreasing due to recovery phenomena [28]. Under low-temperature conditions of Europa this phenomena should be insignificant due to low rate or full stop of metabolism [29].” – section 3.1.
Regarding viability of microorganisms after 10000 years of cryoconservation, it should not be challenging per se (without irradiation), since living cells are found in ancient permafrost rocks not melted for several Myr (e.g. Gilichinsky et al., 1992; Gilichinsky, 2002; Vorobyova et al., 1997). Appropriate comment is added in Introduction (lines 39-41): “Viability of microorganisms after millions of years of cryoconservation should not be challenging per se, since living cells are found in ancient permafrost rocks not melted for several Myr (e.g. [7-9]), but Europa’s orbit being placed in the radiation belt of Jupiter makes moon’s surface subject to high doses of radiation [10].”
Regarding paragraph 3.2, it was completely rewritten:
“The total of all dose accumulation rates of considered incident particles is presented on Fig. 4. In our calculations resulting dose rates for different particles (Figure A1) were close to that of Paranicas et al. [10] at depths less than 10 cm, but for depths more than 10 cm our modelling demonstrated higher dose accumulation values. Those differences increase with depth up to 10 times at 1 m. This is probably the result of using modern models of secondary particle cascade in our GEANT4 calculations. High-energy protons and heavy ions profiles dramatically decrease with depth, thus their impact on the microorganisms’ survival in the ice layer is significant only for depths less than 1 cm, even taking into account their higher biological efficiency comparing to electrons. Despite the high total dose accumulation rate, the lethal dose does not accumulate in ice layer at 0.1-1 m depths until 1000-10000 years after eruption. For times greater than 10000 years the dose is lethal in the region of interest. For exposure times from 1000 years and further we see a nearly constant (in comparison with 1-year profile) part emerging in the dose profile. This is the effect of considering a static resurfacing rate from [14] in our model. This effect might not be real case, but it has no significant impact on the main goals and conclusions of this work.”
References:
Baumstark-Khan, C.; Facius, R. Life under conditions of ionizing radiation. In Astrobiology: The Quest for the Conditions of Life; Horneck, G., Baumstark-Khan, C., Eds.; Springer: Berlin, Germany, 2002; pp. 261–284, ISBN 978-3-642-59381-9, doi:10.1007/978-3-642-59381-9_18
Cheptsov, V.; Vorobyova, E.; Belov, A.; Pavlov, A.; Tsurkov, D.; Lomasov, V.; Bulat, S. Survivability of soil and permafrost microbial communities after irradiation with accelerated electrons under simulated Martian and open space conditions. Geosciences 2018a, 8, 298, doi: 10.3390/geosciences8080298.
Cheptsov, V.S.; Vorobyova, E.A.; Osipov, G.A.; Manucharova, N.A.; Polyanskaya, L.M.; Gorlenko, M.V.; Pavlov, A.K.; Rosanova, M.S.; Lomasov, V.N. Microbial activity in Martian analog soils after ionizing radiation: implications for the preservation of subsurface life on Mars. AIMS Microbiol. 2018b, 4, 541–562, doi:10.3934/microbiol.2018.3.541.
Dartnell, L.R.; Hunter, S.J.; Lovell, K.V.; Coates, A.J.; Ward, J.M. Low-temperature ionizing radiation resistance of Deinococcus radiodurans and Antarctic Dry Valley bacteria. Astrobiology 2010, 10, 717–732, doi:10.1089/ast.2009.0439.
Gilichinsky, D. Permafrost as a microbial habitat. In Encyclopedia of Environmental Microbiology; Bitton, G., Eds.; Wiley: New York, NY, USA, 2002; pp. 932–956, ISBN 978-0-471-35450-5.
Gilichinsky, D.A.; Vorobyova, E.A.; Erokhina, L.G.; Fyordorov-Dayvdov, D.G.; Chaikovskaya, N.R. Long-term preservation of microbial ecosystems in permafrost. Adv. Space Res. 1992, 12, 255–263, doi:10.1016/0273-1177(92)90180-6.
Rummel, J.D.; Beaty, D.W.; Jones, M.A.; Bakermans, C.; Barlow, N.G.; Boston, P.J.; Chevrier, V.F.; Clark, B.C.; de Vera, J.-P.P.; Gough, R.V.; et al. A new analysis of Mars “special regions”: Findings of the second MEPAG Special Regions Science Analysis Group (SR-SAG2). Astrobiology 2014, 14, 887–968, doi:10.1089/ast.2014.1227.
Vorobyova, E.; Soina, V.; Gorlenko, M.; Minkovskaya, N.; Zalinova, N.; Mamukelashvili, A.; Gilichinsky, D.; Rivkina, E.; Vishnivetskaya, T. The deep cold biosphere: Facts and hypothesis. FEMS Microbiol. Rev. 1997, 20, 277–290, doi:10.1111/j.1574-6976.1997.tb00314.x.
Round 2
Reviewer 1 Report
Dear Authors,
You included my corrections and explain why you did not accept others. I'm glad to see corrected version. You improved the text and methods very nice. I have only concerns about the viability and long-term persistence of bacteria on the moon surface. We can not undoubtedly say it. I suggest underlining it is more hypothetical phenomena. After this minor change paper is ready.
Good luck!
Reviewer
Author Response
Dear Reviewer,
Thank you for your kind comments!
During previous round of revision we have added some references (in Introduction) regarding preservation of microorganisms in permafrost during several Myr. So, long-term persistence of bacteria should not be challenging per se (without irradiation). Accumulation of radiation dose should be limiting factor. We have evaluated bacteria viability after irradiation and have calculated radiation dose rates in the Europa ice and based our assumptions on these data. So, we don’t see significant contradictions in our hypothesis. At the same time we understand your doubts. There is could be some overlooked factors (as example - saltation in Mars conditions, which was studied only recently; doi: 10.1089/ast.2018.1856), or Europa life if it exists could have significant differences with Earth life, etc. So, we are downgraded some our assumptions to more hypothetical, and we are added a comment, that we consider ionizing radiation like limiting factor. Also we have underlined, that we studied bacteria survivability under model conditions (meaning it is not fully reproduce real conditions). Please find detailed changes below in bold:
“Combining our numerical modelling of accumulated dose in water ice with observations of water eruption events on Europa we hypothesize that in case of such events it is possible that putative extraterrestrial organisms might retain viability in a dormant state up to 10000 years and be sampled and studied by future probe missions.” – lines 20-23.
“Our study demonstrates the ability of terrestrial microorganisms to survive high intensity radiation in conditions modelling shallow subsurface environments of Europa for a long time.” – lines 175-176.
“Considering radiation dose accumulation like limiting factor for microbial life survivability, we hypothesize that there is a chance to discover the viable cells at ice depth 10-100 cm, if the massive water release from ocean of Europa occurred on a landing site 1000-10000 years before.” – lines 178-181.
Sincerely,
Authors.